# Refinements on the Complementary PDB Construction Mechanism

**Anonymous**

## Abstract

Pattern database (PDB) is one of the most popular automated heuristic generation techniques. A PDB maps states in a planning task to abstract states by considering a subset of variables and stores their optimal costs to the abstract goal in a look up table. As the result of the progress made on symbolic search over recent years, symbolic-PDB-based planners achieved impressive results in the International Planning Competition (IPC) 2018. Among them, Complementary 1 (CPC1) tied as the second best planners and the best non-portfolio planners in the cost optimal track, only 2 tasks behind the winner. It uses a combination of different pattern generation algorithms to construct PDBs that are complementary to existing ones. As shown in the post contest experiments, there is room for improvement. In this paper, we would like to present our work on refining the PDB construction mechanism of CPC1. By testing on IPC 2018 benchmarks, the results show that a significant improvement is made on our modified planner over the original version.

## Introduction

Over past few decades, there has been research on various automated heuristic generation techniques, including abstractions (e.g. PDBs), delete relaxations (e.g. $h^{max}$) and landmarks (e.g. LM-cut) (Helmert and Domshlak 2009). Pattern database (PDB) is a memory-based heuristic generation technique that requires a pre-processing phase. It maps original states to abstract states by considering a subset of variables, namely a pattern, and stores their optimal costs to the abstract goal in a look up table. Early experiments on PDB heuristic search involve optimally solving the sliding-tile puzzles (Culberson and Schaeffer 1998) and the Rubik's cube (Korf 1997). The automated PDB generation process dates back to the work of Edelkamp (2001). Studies have shown that the heuristic could be improved by combining multiple PDBs. Felner, Korf, and Hanan (2004) show that the heuristic from the addition of multiple disjoint PDBs is admissible. On top of that, Holte et al. (2006) suggest taking the maximum heuristic over different additive PDBs. In addition, different representations for PDBs have been proposed. Explicit PDBs are structured as hash tables and usually limited by memory, while symbolic PDBs (Edelkamp 2002) are structured as binary decision diagrams (BDDs)

(Bryant 1986). The succinct representation of state sets in BDDs usually reduces the memory consumption and enables the construction of much larger PDBs.

The space of possible abstractions is huge due to exponentially many patterns and various cost-partitioning strategies. In order to construct PDBs with good quality, several algorithms have been proposed, including genetic algorithm (Edelkamp 2006), bin-packing (Edelkamp 2006; Franco et al. 2017) and hill climbing (Haslum et al. 2007; Kissmann and Edelkamp 2011). Also, some PDB evaluation approaches have been suggested, such as average heuristic value (Edelkamp 2006), random walk sampling (Haslum et al. 2007) and stratified sampling (Lelis et al. 2016; Barley, Franco, and Riddle 2014). The basic idea of the Complementary PDB construction (CPC) mechanism is to keep generating PDB collections using a combination of the above-mentioned algorithms and add a new collection $\mathcal{P}_{sel}$ to the collection set $\mathcal{S}_{cur}$ if $\mathcal{S}_{cur} \cup \{\mathcal{P}_{sel}\}$ is predicted to improve the heuristic. Moreover, the parameters of the PDB construction algorithms are dynamically adjusted in the course.

The results of IPC 2018 have confirmed the effectiveness the CPC mechanism. Both Complementary 1 (CPC1) and Complementary 2 (CPC2) planners solved 124 planning tasks and tied as the runner ups, only 2 tasks behind the winner, and they were the best non-portfolio planners. Nonetheless, some ill behaviors of CPC1 have been revealed in the post contest experiments (Franco et al. 2018). For example, using a single PDB construction algorithm, GAMER-Style (Kissmann and Edelkamp 2011), can solve substantially more tasks than using a combination of algorithms, which indicates that these algorithms are not well integrated. Hence, the purpose of our research is to refine the PDB construction mechanism of CPC1 by optimizing the algorithms and the evaluator as well as better integrating all components.

In this paper, the definitions of related concepts will be introduced. Then, our refinements on the pattern construction algorithms, the evaluator and the overall process will be presented. Finally, the results from the experiments on IPC 2018 benchmarks will be summarized to demonstrate the effectiveness of our refinements.

## Background

### SAS+ Planning

Planning tasks are encoded into the SAS+ model on the Fast Downward planning system (Helmert 2006). An SAS+ planning task is defined by a 4-tuple $\Pi = \langle \mathcal{V}, \mathcal{O}, s_0, s_* \rangle$ (Bäckström and Nebel 1995). $\mathcal{V}$ is a finite set of state variables, where each variable $v \in \mathcal{V}$ has as finite domain $\mathcal{D}_v$. A full state is an assignment to all state variables, while a partial state is an assignment to a subset of state variables. $\mathcal{O}$ is a finite set of operators, where each operator $o \in \mathcal{O}$ is defined by a tuple $\langle pre_o, eff_o, cost_o \rangle$, specifying the preconditions, the effects and the non-negative cost. Both $pre_o$ and $eff_o$ are partial states. An operator $o$ is applicable to a state $s$ if and only if $s$ agrees on the values of all variables in $pre_o$. The result of applying $o$ to $s$ is a new state $s'$ that agrees on the values of all variables in $eff_o$ with the other variables unchanged. $s_0$ is the initial state and $s_*$ is the partial goal state. A solution to a planning task is a defined sequence of operators $(o_1, \ldots, o_n)$ that leads from $s_0$ to a goal state that satisfies $s_*$, i.e. $o_n(\ldots o_1(s_0) \ldots)[v] = s_*[v]$ for all $v \in Var(s_*)$. The task of cost optimal planning is to find a solution with the least cost.

The causal graph of an SAS+ planning task is a directed graph $(V, E)$ with $V = \mathcal{V}$ and $(u, v) \in E$ if and only if there is an operator $o \in \mathcal{O}$ such that $eff_o(v)$ is defined and either $pre_o(u)$ or $eff_o(u)$ is defined, $u \neq v$ (Helmert 2004). An arc $(u, v)$ in the causal graph implies that the change of $v$ is dependent on the current assignment of $u$.

### Pattern Databases

A pattern database (PDB) is a look up table of abstract states that stores their optimal costs to the goal. It abstracts the problem space by mapping it to a subset of state variables $\mathcal{P} \subseteq \mathcal{V}$, i.e. a pattern, and removing from the preconditions and effects of operators, $s_0$ and $s_*$ any variable not in $\mathcal{P}$. This kind of abstraction is said to be homomorphic as any transition $(s, s')$ in the original problem space remains valid, thereby yielding admissible and consistent heuristic. A PDB is constructed by a blind regression search from the abstract goal states (Culberson and Schaeffer 1998; Edelkamp 2001), usually up to a time and memory limit. If the limit is reached and the search is stopped at the depth $d$, a partial PDB will be formed and a heuristic value of $d$ plus the minimum operator cost will be returned for any unvisited abstract state (Anderson, Holte, and Schaeffer 2007). The size of a PDB is defined as $\Pi_{v \in \mathcal{P}} |\mathcal{D}_v|$, which is the cross product of the domain size of all variables in $\mathcal{P}$.

A combination of multiple PDBs generally produces better heuristic, e.g. through addition (Felner, Korf, and Hanan 2004) and maximization (Holte et al. 2006). Multiple PDBs are additive if any operator $o \in \mathcal{O}$ affects at most only one PDB, where $o$ affects a PDB $\mathcal{P}$ if $Var(eff_o) \cap \mathcal{P} \neq \emptyset$. Given a PDB collection, the additivity can be ensured with cost-partitioning, which splits the operator costs among the PDBs. A simple approach adopted in the Complementary planners, zero-one cost- partitioning, is to set the cost of an operator $o$ to zero if it has affected any PDB in the collection, and keep the original cost otherwise (Edelkamp 2001). There are alternatives that could yield more informative heuristic, such as saturated cost-partitioning (Seipp et al. 2017), but they are usually more computationally expensive. All the PDB collections are combined with the canonical function (Haslum et al. 2007) in the Complementary planners, which takes the maximum heuristic value over all admissible combinations.

### Symbolic Search and PDBs

Symbolic search has gained tremendous popularity in cost optimal planning because of significant memory saving and faster computation. A state vector is a binary encoding of the variables and its length is given by $\sum_{v \in \mathcal{V}} \lceil log |\mathcal{D}_v| \rceil$. A characteristic function is used to represent a set of states that returns true if and only if an encoded state is in the set. In symbolic search, any characteristic function is succinctly represented in a binary decision diagram (BBD) (Bryant 1986). In addition, any operator can be represented as a set of transition relations $T_c(s, s')$ in a BBD, where $c$ is the operator cost, $s$ is the predecessor state and $s'$ is the successor state. The set of successors or predecessors for a state set can be generated via the image or the preimage operation, which takes the relational product of the state set and the transition relations (Torralba et al. 2017). The size of a BBD depends on the ordering of the variables. A local search algorithm has been proposed by Kissmann and Edelkamp (2011) that places related variables in the causal graph as close as possible, which performs much better than random ordering.

Symbolic A* search (BBDA*) partitions the open list into a matrix of g and h value where each bucket is represented by a BBD (Edelkamp and Reffel 1998; Jensen, Bryant, and Veloso 2002). It expands the buckets along an f diagonal each time, starting from the lowest g value, and adds the successor state sets, to avoid duplicate expansion. Note that in the presence of zero-cost operators, additional layers in a bucket are needed to keep track of the blind search using only those operators. Symbolic PDBs (Edelkamp 2002) can be constructed in a way similar to explicit PDBs. A regression search is made from the set of abstract goals, and the sets of expanded abstract states are encoded in BBDs along with their costs to goal. It is convenient to query on a heuristic value for a state set by doing a conjunction with the BBDs, producing a subset of states with the heuristic value.

## The Complementary PDB Construction Mechanism

### GAMER-Style

GAMER-Style (Kissmann and Edelkamp 2011) is a hill climbing algorithm for constructing a single large PDB. The sketch is given in Algorithm 1. It starts with all goal variables and sequentially adds a set of causally related variables that are predicted to improve the heuristic. If the set of selected variables is empty, GAMER-Style will either terminate if the candidate variable set is also empty, or continue with the remaining candidate variables in the next call, i.e. a partial GAMER run. In our implementation, the remaining candidate variables are shuffled before the partial run to avoid the effect of ordering (line 6).

The performance of GAMER-Style depends a lot on the evaluator $E$. Although originally average h is used, random walk sampling is suggested by Franco and Torralba (2019). Compared to average h, random walk sampling gives more precise estimate because the problem space distribution is approximated and known dead ends are removed (Haslum et al. 2007). In light of this, our evaluator samples separately for GAMER-Style according to the initial h of $\mathcal{P}_{sel}$. Resampling will be performed if and only if the initial h of the new PDB increases by over 10% (line 16). More details on the evaluator are explained in the Evaluator section.

---

**Algorithm 1:** GAMER-Style

**Input** : Planning task $\Pi$, evaluator $E$, time limit $T$ and memory limit $M$

1 **if** *called first time* **then**
2    $\mathcal{P}_{sel} \leftarrow$ goal variables in $\Pi$;
3 **if** $\mathcal{S}_{can} = \emptyset$ **then**
4    $\mathcal{S}_{can} \leftarrow$ variables causally related to $\mathcal{P}_{sel}$;
5 **else**
6    Shuffle $\mathcal{S}_{can}$;
7 threshold$\leftarrow E(\mathcal{P}_{sel})$;
8 **while** $\mathcal{S}_{can} \neq \emptyset$ *and iteration time*$< 120s$ *and* $t < T$ *and* $m < M$ **do**
9    candidate variable$\leftarrow \mathcal{S}_{can}$ back, $\mathcal{S}_{can}$ pop back;
10    $\mathcal{P}_{can} \leftarrow \mathcal{P}_{sel} \cup \{$candidate variable$\}$;
11    candidate value$\leftarrow E(\mathcal{P}_{can})$;
12 **if** *the highest candidate value is above the threshold* **then**
13    $\mathcal{S}_{sel} \leftarrow$ candidate variables whose value is within 0.1% margin of the highest candidate value;
14 **if** $\mathcal{S}_{sel} \neq \emptyset$ **then**
15    $\mathcal{P}_{sel} \leftarrow \mathcal{P}_{sel} \cup \mathcal{S}_{sel}$;
16    $E$ resamples if initial h of $\mathcal{P}_{sel}$ rises by over 10%;
17    $\mathcal{S}_{can} \leftarrow \emptyset$;    // candidate variables regenerated in the next call
18    return $\mathcal{P}_{sel}$;
19 **else**
20    **if** $\mathcal{S}_{can} = \emptyset$ **then**
21      GAMER-Style terminates;

---

## Bin-Packing

Next-Fit Bin-packing (Moraru et al. 2019) consists of two algorithms, Next-Fit Decreasing Bin-Packing (NFD) and Next-Fit Increasing Bin-Packing (NFI), which differ in the order of candidate variables $O$. The sketch is given in Algorithm 2. It has been shown empirically that Next-Fit Bin-Packing is effective for seeding (Franco et al. 2018). In our implementation, an additional step of shuffling the causally related variables is added to inject randomness (line 11).

Causal Dependency Bin-Packing (CBP) (Franco et al. 2017), as sketched in Algorithm 3, is used in the main PDB construction phase. Each pattern contains $N$ randomly selected goal variables and iteratively adds causally related variables. The PDB collection is sorted in the descending order with respect to the pattern length before being returned (line 14). Compared to Next-Fit Bin-Packing, CBP is more random because candidate variable are not sorted and $N$ can be adjusted. Both bin-packing algorithms are implemented with more efficient data structures in CPC0.

---

**Algorithm 2:** Next-Fit Bin-Packing

**Input** : Planning task $\Pi$, sorting order $O$ and size limit $S$

1 $\mathcal{S}_{can} \leftarrow$ variables in $\Pi$ whose domain size is below $S$;
2 Sort $\mathcal{S}_{can}$ according to $O$;
3 $\mathcal{P}_{sel} \leftarrow \emptyset, P \leftarrow \emptyset$;
4 **while** $\mathcal{S}_{can} \neq \emptyset$ **do**
5    $v \leftarrow \mathcal{S}_{can}$ front, $\mathcal{S}_{can}$ pop front;
6    **if** *P has no more space for v* **then**
7      $\mathcal{P}_{sel} \leftarrow \mathcal{P}_{sel} \cup \{P\}$;
8      $P \leftarrow \emptyset$;      // Restart a new bin
9    $P \leftarrow P \cup \{v\}$;
10    $\mathcal{S}_{rel} \leftarrow$ variables in $\mathcal{S}_{can}$ causally related to $v$;
11    Shuffle $\mathcal{S}_{rel}$;
12    **while** $\mathcal{S}_{rel} \neq \emptyset$ **do**
13      $v' \leftarrow \mathcal{S}_{rel}$ front, $\mathcal{S}_{rel}$ pop front;
14      **if** *P has space for v'* **then**
15        $P \leftarrow P \cup \{v'\}$, remove $v'$ from $\mathcal{S}_{can}$;
16 $\mathcal{P}_{sel} \leftarrow \mathcal{P}_{sel} \cup \{P\}$;    // Add the last bin
17 return $\mathcal{P}_{sel}$;

---

## Evaluator

The evaluator plays a crucial role in the PDB construction process as it decides which PDB collections to be included in the set $\mathcal{S}_{cur}$. The quality of a PDB collection is usually evaluated with respect to the reduction on the search tree size or the search time. Since the search tree size is negatively related to the average heuristic value as conjected by Korf (1997), the heuristic value is used as the metric for evaluators such as average h (Edelkamp 2006) and random walk sampling (Haslum et al. 2007). A stratified-sampling-based evaluator (Lelis et al. 2016; Barley, Franco, and Riddle 2014) that takes the predicted search time as the metric is employed in CPC2 (Franco, Lelis, and Barley 2018), whereas random walk sampling evaluator is used in CPC1. As mentioned in (Franco et al. 2017), both evaluators yield similar results on symbolic PDBs. An explanation is that since a symbolic PDB collection generally contains fewer PDBs than an explicit one, the look up time of a symbolic PDB collection does not vary that much, resulting in a stronger correlation between the search tree size and the search time. For faster evaluation, random walk sampling evaluator is also adopted in CPC0 with some modifications and extensions.

The random walk sampling evaluator samples each state by applying a number of uniformly chosen operators according to the heuristic value of the initial state. After each step, if the state is known to be a dead, the random walk returns

**Algorithm 3:** Causal Dependency Bin-Packing

**Input** : Planning task $\Pi$, number of goal variables to place $N$ and size limit $S$

1   $\mathcal{S}_{can} \leftarrow$ variables in $\Pi$ whose domain size is below $S$;
2   $\mathcal{S}_{can\_g} \leftarrow$ goal variables in $\Pi$ whose domain size is below $S$;
3   $\mathcal{P}_{sel} \leftarrow \emptyset$, $P \leftarrow \emptyset$;
4   **while** $\mathcal{S}_{can\_g} \neq \emptyset$ **do**
5      $\mathcal{S}_{sel\_g} \leftarrow N$ variables randomly taken from $\mathcal{S}_{can\_g}$, remove them from $\mathcal{S}_{can}$ and $\mathcal{S}_{can\_g}$;
6      $P \leftarrow P \cup \mathcal{S}_{sel\_g}$;
7      $\mathcal{S}_{rel} \leftarrow$ variables in $\mathcal{S}_{can}$ causally related to $\mathcal{S}_{sel\_g}$, shuffle $\mathcal{S}_{rel}$;
8      **while** *there is $v \in \mathcal{S}_{rel}$ that fits into $P$* **do**
9         $P \leftarrow P \cup \{v\}$, remove $v$ from $\mathcal{S}_{rel}$, $\mathcal{S}_{can}$ and $\mathcal{S}_{can\_g}$;
10        $\mathcal{S}_{rel'} \leftarrow$ variables in $\mathcal{S}_{can}$ causally related to $v$;
11        $\mathcal{S}_{rel} \leftarrow \mathcal{S}_{rel} \cup \mathcal{S}_{rel'}$, shuffle $\mathcal{S}_{rel}$;
12      $\mathcal{P}_{sel} \leftarrow \mathcal{P}_{sel} \cup \{P\}$;
13      $P \leftarrow \emptyset$;      // Restart a new bin
14 Sort $\mathcal{P}_{sel}$ in descending order with respect to the pattern length;
15 **return** $\mathcal{P}_{sel}$;

to the initial state. In our evaluator, the sampling stops when either the 10,000 state size limit or the 30s time limit is reached, and unique states are taken from the sample. In case the sample is empty when all sampled states are dead ends, the initial state is always included. Then, the heuristic value of $\mathcal{S}_{cur}$ is stored for each sample state. To evaluate a new PDB collection $\mathcal{P}_{sel}$, the heuristic value of $\mathcal{P}_{sel}$ is compared with with the stored value on each sample state, and true is returned if and only if at least 25% states are improved. The evaluation criteria can be summarized by

$$\sum_{i=1}^{m} \frac{h_{\mathcal{P}_{sel}}(s_i) > h_{\mathcal{S}_{cur}}(s_i)}{m} \geq 0.25$$

where m is the sample size and $\{s_1, \ldots, s_m\}$ is the set of sample states. Compared to the hard threshold in the original evaluator, the ratio threshold is more accurate and flexible for varying sample size. After each evaluation, new dead ends are removed from sample state, and the heuristic value of each sample state is updated if $\mathcal{P}_{sel}$ is included. Resampling will be performed if and only if the heuristic value of the initial state increases by over 10% to save time.

The time overhead in maximizing over a set of PDB collections has been addressed by researchers (Holte et al. 2004; Barley, Franco, and Riddle 2014). To reduce the size of $\mathcal{S}_{cur}$ while maintaining the quality, the pruning of dominated collections is introduced. The pruning progresses backwards from the last PDB collection and stores the maximum heuristic value of processed PDB collections for each sample state. A PDB collection is pruned if and only if there is no state where its heuristic value exceeds that of any previous collection. The pruning order is determined as such

because earlier added PDB collections are more likely to be dominated by later ones. This strategy is safe since it is quite rare for a useful PDB collection to underperform on every sample state.

## Adaptive PDB Construction

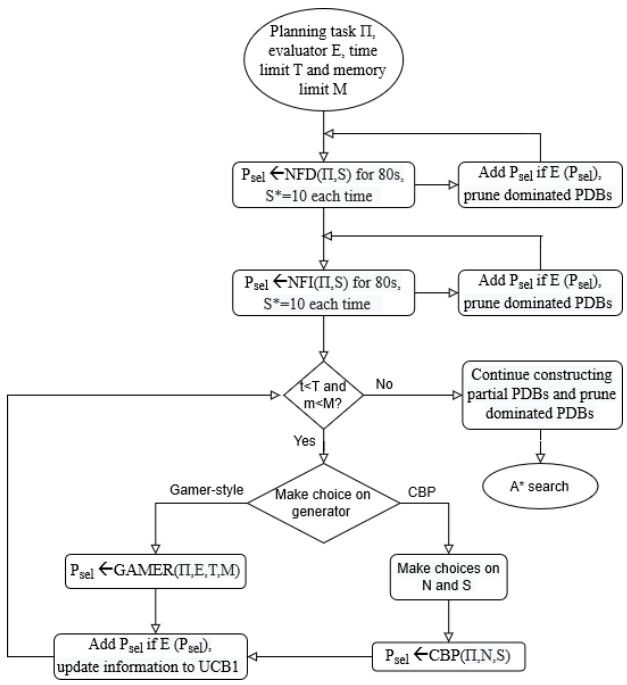

Figure 1: The PDB construction process

The CPC process, as illustrated in Figure 1, starts with NFD and then NFI. Both of them run for 80s excluding the resampling time and start with a size limit of $10^8$ that is scaled by 10 after each iteration. Whenever a new PDB collection is added, resampling will be performed if the initial h rises by over 10%, and dominated PDB collections will be pruned. The pruning is reasonable since the PDB collections formed in this phase are homogeneous and the time overhead for subsequent evaluations will be reduced. The largest size limit where a PDB is added is recorded as $S_l$ for later use. In CPC1, the perimeter PDB is constructed prior to the bin packing. However, since the perimeter PDB usually takes too much time and memory to be useful and may disadvantage other algorithms, it is not adopted in CPC0.

After the seeding phase, the PDB construction process goes on until the time or the memory limit is reached. At each iteration, a choice is made between CBP and GAMER-Style using the UCB1 formula, which is a learning policy that balances exploration versus exploitation in a multi-armed bandit problem and uniformly approaches the minimal expected cumulative regret (Auer, Cesa-Bianchi, and Fischer 2002). The score for each choice is calculated by $\bar{x}_i + \sqrt{\frac{2\ln n}{n_i}}$, where $\bar{x}_i$ is the average reward received by the choice $i$, $n_i$ is the number of times $i$ is chosen and $n$ is the total number of trials, and the choice with the highest score

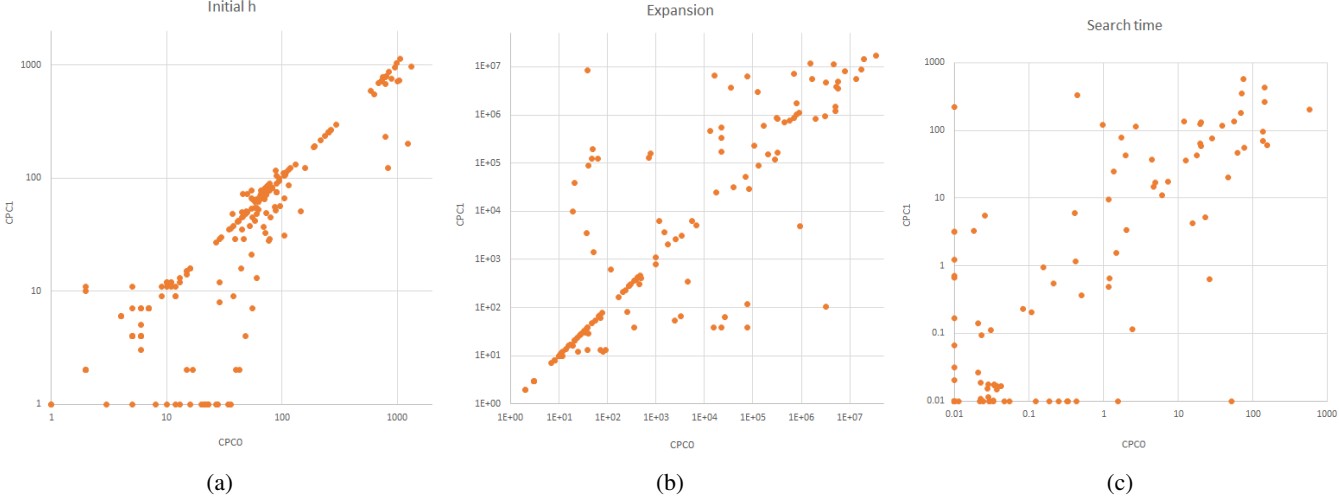

Figure 2: Comparison between CPC0 and CPC1 on the planning tasks

is selected each time. In our planner, $n_i$ and $n$ are updated with the PDB construction time of the algorithm $i$, and the reward of $i$ is also increased if the PDB is added to $\mathcal{S}_{sel}$. The formula used in our planner is revised on the second term to the correct form so that it will not be biased to explored choices.

For CBP, the parameters $N$ and $S$ are decided using UCB1 as well. The choices for $N$ range from 1 to the number of goal variables in $\Pi$ and the choices for $S$ are drawn from a set of sizes ranging from $10^9$ to $10^{35}$, with the ones over $10^4 * S_l$ filtered out to avoid PDBs too large to construct. The maximum size in CPC1 is $10^{20}$. GAMER-Style complements the Bin-Packings as it constructs a single PDB that contains all goal variables, whereas the Bin-Packings construct a PDB collection with goal variables distributed among them. By integrating these algorithms, the adaptive PDB construction process can generate a great variety of PDBs and learn the most suitable algorithm to use as well as the best configuration. Before the A* search, the construction of any partial PDB will continue if there is excess time and memory and dominated PDB collections will be pruned.

## Experiments

Both CPC0 and CPC1 are built upon an early 2017 fork of the Fast Downward planning system (Helmert 2006) with the $h^2$-based pre-processor to remove irrelevant operators (Alcázar and Torralba 2015). The symbolic search is enhanced with the techniques proposed by Torralba et al. (2017). The optimal STRIPS benchmarks from IPC 2018 are used for the evaluation, consisting of 12 domains (split ones included) and 240 planning tasks (20 tasks in each domain). All experiments are conducted on an Intel i5-6200U 2.30GHz machine, under the IPC 2018 rule of 30-minute time limit and 8 GB memory limit. The PDB construction is set to the time limit at 15 minutes and the memory limit at 4 GB for CPC1. Since PDB construction is the most important stage in our planners and effective PDBs may significantly

reduce the search efforts, the time limit is raised to 18 minutes for CPC0.

## Overall Results

The key statistics from our experiments are summarized in Table 1, Table 2 and Figure 2. The initial h values are taken from the tasks where they are available in both planners. The expansions, the search time, the total time and the memory usage are taken from the tasks solved by both planners. It can be seen from the tables and Figure 2(a) that CPC0 usually produces PDBs with higher initial h value. This indicates that the refined algorithms and evaluator can generate better heuristic. While Table 2 shows that the number of tasks where less nodes are expanded in CPC0 is almost the same as the number of tasks where more nodes are expanded, Table 1 shows that CPC0 has slightly fewer node expansions on average. Looking at Figure 2(b), the instances on CPC1 side lie further off the diagonal than those on CP0 side. Moreover, CPC1 tends to expand more nodes for the tasks requiring a great number of expansions. It is because the refined process in CPC0 is able to construct PDBs of consistent quality over tasks of varying difficulty. According to Table 2 and Figure 2(c), the search time of CPC0 is generally less than that of CPC1, and the average search time of CPC0 is significantly reduced, as per Table 1. This may attribute to the pruning of dominated PDBs that reduces the look up time. A combination of better heuristic and faster heuristic look up enables CPC0 to solve 12 more tasks. Hence, our refinements make the planner even more competitive in the context of IPC 2018.

On the other hand, the total time and the memory usage of CPC0 are increased on average as per Table 1. It is because we have raised the time limit for PDB construction. Also, the drop of perimeter PDB reduces the chance of solving a task during PDB construction, thereby increasing the total time and the memory usage for solved tasks.

| Domain | CPC0 | | | | | | CPC1 | | | | | |
|---|---|---|---|---|---|---|---|---|---|---|---|---|
| | Coverage | Avg init h | Avg expansions | Avg search time(s) | Avg total time(s) | Avg memory(KB) | Coverage | Avg init h | Avg expansions | Avg search time(s) | Avg total time(s) | Avg memory(KB) |
| Agr | 13 | 871.90 | 3733913.90 | 96.58 | 1217.22 | 3593952.00 | 10 | 705.00 | 3281583.80 | 74.14 | 901.70 | 3861196.40 |
| Cal | 12 | 11.00 | 6190.45 | 0.53 | 1120.37 | 3155986.18 | 11 | 9.15 | 97475.45 | 19.60 | 328.07 | 1982259.27 |
| Cal-S | 10 | 52.95 | 92597.11 | 2.52 | 1096.14 | 3904275.11 | 9 | 51.05 | 794227.33 | 14.67 | 244.18 | 1732187.56 |
| DN | 14 | 80.90 | 166718.07 | 7.32 | 784.22 | 2426417.43 | 14 | 60.00 | 977729.86 | 30.74 | 616.50 | 2668070.00 |
| Nur | 16 | 42.10 | 786155.14 | 11.38 | 738.95 | 2462988.86 | 14 | 32.25 | 1187245.71 | 57.83 | 464.78 | 2465164.86 |
| OSS | 7 | 1.71 | 2.71 | 0.01 | 7.16 | 924884.00 | 7 | 1.71 | 2.71 | 0.01 | 6.64 | 924018.29 |
| OSS-S | 13 | 141.00 | 79405.58 | 3.38 | 760.81 | 2343938.67 | 13 | 140.83 | 437.08 | 0.06 | 189.66 | 1099061.33 |
| PNA | 20 | 152.11 | 8917.44 | 0.32 | 840.22 | 2157075.17 | 18 | 147.50 | 567690.61 | 20.59 | 311.86 | 2009214.67 |
| Set | 10 | 57.67 | 750204.44 | 20.95 | 1184.12 | 3146579.11 | 9 | 53.33 | 586050.44 | 79.82 | 1155.60 | 3501106.22 |
| Sna | 14 | 22.25 | 197577.64 | 4.74 | 1202.88 | 4629628.18 | 11 | 2.70 | 1804478.00 | 45.88 | 872.29 | 5134996.00 |
| Spi | 11 | 3.75 | 824340.18 | 25.19 | 1227.99 | 4099398.18 | 12 | 4.40 | 428587.09 | 24.95 | 1158.53 | 3477634.55 |
| Ter | 16 | 76.70 | 4834400.00 | 18.84 | 965.54 | 2871772.75 | 16 | 81.65 | 2498460.25 | 9.78 | 751.06 | 2656157.25 |
| Average | 13 | 135.55 | 1042518.09 | 14.84 | 935.46 | 2943388.75 | 12 | 114.55 | 1066077.47 | 30.66 | 582.07 | 2624802.31 |

Table 1: Average statistics of CPC0 and CPC1 on IPC 2018 benchmarks

| Domain | Coverage | Higher init h | Fewer expansions | Less search time |
|---|---|---|---|---|
| Agr | 3 | 4 | -4 | 0 |
| Cal | 1 | 0 | -1 | 3 |
| Cal-S | 1 | -3 | -3 | 0 |
| DN | 0 | 10 | 3 | 2 |
| Nur | 2 | 6 | 2 | 5 |
| OSS | 0 | 0 | 0 | 0 |
| OSS-S | 0 | 1 | -4 | -1 |
| PNA | 2 | 5 | 1 | 3 |
| Set | 1 | 3 | -3 | 6 |
| Sna | 3 | 20 | 11 | 8 |
| Spi | -1 | 3 | 5 | 1 |
| Ter | 0 | -14 | -10 | -5 |
| Total | 12 | 35 | -3 | 22 |

Table 2: Comparison between CPC0 and CPC1 on each domain. In each cell is the number of tasks where CPC0 outperforms CPC1 minus the number of tasks where CPC1 outperforms CPC0. The search time of a task is counted if the difference is greater than 1s.

|  | Domain | Nur | Sna |
|---|---|---|---|
| CPC0 | Coverage | 16 | 14 |
| | Avg init h | 43.60 | 27.55 |
| | Avg expansions | 1741837.81 | 13672.93 |
| | Avg search time(s) | 26.22 | 0.36 |
| | Avg total time(s) | 626.02 | 893.16 |
| | Avg memory(KB) | 2214411.75 | 3944160.00 |
| CPC1 | Coverage | 16 | 14 |
| | Avg init h | 42.95 | 21.05 |
| | Avg expansions | 1607741.94 | 832964.36 |
| | Avg search time(s) | 43.82 | 28.78 |
| | Avg total time(s) | 622.18 | 1053.54 |
| | Avg memory(KB) | 2418533.00 | 3744610.29 |
| Comparison | Higher init h | 5 | 6 |
| | Fewer expansions | 3 | 3 |
| | Less search time | 5 | 5 |

Table 3: Statistics of two versions of GAMER-Style

## GAMER-Style Comparison

In CPC0, the variable selection method of GAMER-Style is changed from average h value to random walk sampling. To evaluate the effect, both versions are tested on Nurikabe and Snake, where GAMER-Style contributes the most. In our experiments, other PDB construction algorithms are disabled, and both planners are set to the time limit at 900s and the memory limit at 4GB for PDB construction and are tested under the IPC 2018 rule.

It can be seen from Table 3 that the new version outperforms the original version in Snake. Despite a slight increase in the average memory usage, there is a significant improvement in the other statistics, especially a huge reduction in the average node expansions. The difference is not that obvious in Nurikabe. Although the new version has slightly more node expansions on average, mainly due to the task

14, it has fewer expansions on 3 more tasks. Also, the average initial h value of the new version is higher, and the initial h value is higher on 5 more tasks. Thus, the new version is more likely to outperform the original version in Nurikabe. In both domains, the average search time of the new version is significantly reduced, due to not only better heuristic, but also the pruning method. Since the PDBs constructed by GAMER-Style are homogeneous, it is very beneficial to prune the dominated ones.

## Related Work

GA-PDB (Edelkamp 2006) uses the the bin-packing algorithm and the genetic algorithm to construct PDB collections, which are evaluated by average heuristic value. iPDB (Haslum et al. 2007) starts with a collection of PDBs each containing a goal variable and in each iteration constructs a new PDB by adding a causally related variable to an existing PDB. The PDBs are evaluated by random walk sampling and combined with the canonical function. GAMER (Kissmann and Edelkamp 2011) performs symbolic A* search with the symbolic PDB constructed by GAMER-style algorithm. It has an alternative option to perform symbolic bidirectional blind search, depending on the property of the planning task.

CPC2 (Franco, Lelis, and Barley 2018) is an early version of the Complementary planers. The difference is that CPC2 does not perform Next-Fit Bin-Packing or GAMER-Style but uses CBP and Regular Bin-Packing (RBP) for PDB construction and performs random mutations to selected PDB collections. The PDB size limits are drawn from the binomial distribution with the parameters dynamically adjusted. A stratified-sampling-based evaluator is used, which evaluates the PDB with respect to the predicted search time and adds a PDB collection if the search time is predicted to be reduced. The pruning of dominated PDB collections is performed after every certain time period.

## Conclusions

In this paper, we have presented our refinements on the complementary PDB construction mechanism, including GAMER-Style, Next-Fit Bin-Packing, Causal Dependency Bin-Packing, as well as the random walk sampling evaluator. We have optimised the algorithms, extended the evaluator and improved the overall process. Our experiments show that these algorithms has been well integrated and the modified planner CPC0 has a significantly higher coverage on IPC 2018 benchmarks than the original planner CPC1. Therefore, the effectiveness of the CPC mechanism in cost optimal planning is again confirmed and a competitive benchmark planner is contributed.

For future research, some other PDB construction algorithms and cost-partitioning strategies, e.g. saturated cost-partitioning (Seipp et al. 2017), could be introduced. Also, more advanced evaluators such as stratified sampling (Lelis et al. 2016; Barley, Franco, and Riddle 2014) may improve the PDB selection process. A major drawback of PDB-based planners is the need of pre-processing phase, which increases the total time for solving a task. Techniques like interleaved search and heuristic improvement (Franco and Torralba 2019) could be further explored to address this issue.

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
