# OpenReview forum: "Refinements on the Complementary PDB Construction Mechanism"
_icaps-conference.org/ICAPS/2020/Workshop/HSDIP — Submitted to HSDIP 2020_

### Official Review · AnonReviewer2 · 2020-07-31
**This paper suggests some technical improvements for the CPC1 planner but lacks conceptually new ideas and a proper evaluation.**

**Confidence:** 5
**Rating:** 3

**Review:**

This paper describes the CPC1 planner from IPC 18 and attempts at making small
changes to some of its components. Unfortunately, it is not always clear what
exactly those changes are. Also, for those differences that I think I
recognized as change, they are often of technical but not conceptual nature:
shuffling variable candidates to introduce (even more) randomness to a planner
which already greatly depends on random seeds, using "more efficient data
structures" (which are not described at all), resampling taking place only
after a certain threshold (I'm not sure if that was already present in CPC1 or
not), or switching from average to initial h values (again I'm not sure if this
is a difference; it if is, it would be a small conceptual change at least). (On
a side note, I wondered throughout the paper what CPC0 is, only to read in the
conclusion that this is the modified variant of CPC1. I think that this is a
*very* bad choice for a name because it makes the impression that CPC 0 through
2 are all original IPC planners.) However, even if all the details were clear,
I don't think that this investigation of the planner is interesting enough to
warrant being published. I rather feel that these modifications are engineering
tweaks and optimizations that would fit a technical report or a planner
abstract. In fact, this paper reads a lot like an overhauled version of the
planner abstract of CPC1. This feeling is reinforced particularly due to the
paper not introducing any new techniques nor comparing to any other pattern
selection method or PDB combination technique. All in all, this is not enough
for HSDIP.

Besides the lack of distinction from the original IPC planner, the details in
the write-up and, in particular, the pseudo-code, are unclear to the point that
I wouldn't be able to implement the algorithms by just reading this paper. The
algorithms contain lots of magic constants or arbitrary decisions (such as if
and when to resample, pruning PDBs or not, using perimeter PDBs or not, runtime
limits for the "seeding phase", increment of memory, the threshold for the
evaluator value for deciding if a new pattern should be kept or not, and many
more such as percentages, time and memory limits etc.). All of these arbitrary
choices should be parameters of the algorithm and properly evaluated to justify
the decisions and to provide some interesting insights to take away from this
paper.

The pseudo-code also often assumes that certain variables or methods are
globally available but of which no details are given. Furthermore, I think that
pseudo-code could be agnostic of data structures (or it should detail out the
data structure used if that is of importance): it doesn't matter if an element
is taken from the "back" or the "front" (Algorithms 1 and 2) of some set;
mathematically, there is no order. I realize the intention here is that these
sets are rather lists, but then S = \emptyset doesn't make sense. To simplify
the presentation, instead of shuffling some set/list S, it would suffice to
simply choose some element at random from S whenever some element should be
chosen.

Next, coverage of related work should be improved. This paper fails to mention
the following other pattern generation techniques:
- systematic generation of "interesting" patterns (Pommerening et al. IJCAI 2013)
- repeated random walks over the causal graph to compute patterns (Rovner et al.
  ICAPS 2019)
- CEGAR as a more sophisticated algorithm of the above (Rovner et al. ICAPS 2019)
It also doesn't mention the following PDB combination techniques based on cost
partitioning in the introduction/related work (only somewhere within the
paper), even though the intro mentions "various cost-partitioning strategies":
- post-hoc optimization (Pommerening et al. IJCAI 2013)
- saturated cost partitioning (Seipp et al. JAIR 2020)
Also note that the canonical function can be understood as cost-partitioning; I
recommend Seipp et al. ICAPS 2017 for a nice overview of cost partitioning
methods that can all be easily used with PDBs. Also note that perimeter PDBs
are never defined/explained nor cited, I would recommend Eyerich & Helmert
ICAPS 2013.

Finally, as already mentioned, the experimental evaluation should
(individually!) evaluate the design choices which have been arbitrarily fixed,
and in particular, of course, those aspects that changed compared to CPC1 as to
investigate which change has which impact. Also there is no reason to restrict
the benchmarks to IPC 2018 domains. Since the two planners very much depend on
randomization (CPC has a very large variance according to my own experience),
it is mandatory to consider the average over several runs with different random
seeds (ideally 10 or more runs). It is also absolutely mandatory that both
planners use the *same* time limit. Why would one use 3 minutes more than the
other?! The entire evaluation with this difference is void; how can I know if
the "improvements" are not only due to spending more time to compute PDBs?

Some details:

- citing LM-cut for landmarks: while LM-cut uses landmarks, I would consider
the "classical" landmark heuristics to be those considered for satisficing
planning

- LaTeX: $eff$ and $Var$ looks very ugly; one should use $\textit{...}$ for all
text which is used in math mode

- notation o(s) and Var(s) is not defined

- "to find a solution": + "or to prove that no solution exists"

- Algorithm 1: define small t and m; "highest candidate value" ?; S_sel can be
undefined in line 14; "E resamples" ?; "initial h of P_sel": generally, it is
unclear to me what the heuristic value of a current pattern/pattern selection
is -- which heuristic is used for this? The canonical heuristic over the
current candidate/collection? The same question applies to the evaluator E
which uses some heuristic.

- "PDB collection is sorted in descending order": why?

- "main PDB construction phase": at this point, the reader cannot know what
this is

- "CBP is more random": I don't understand, how do you measure randomness? If
anything, from what I understand, CBP is *less* randomized because it
guarantees the inclusion of goal variables and variables relevant to the
already chosen variables. For NFD/NFI, I don't see that guarantee, which means
that even patterns whose PDBs are equivalent to the constant 0-heuristic can be
produced.

- "more efficient data structures in CPC0": this is the first mention of CPC0;
what are more efficient data structures? compared to what?

- "crucial role in the PDB construction process": what is that process?
GAMER-Style? any of the three?

- "S_cur": what is this? it doesn't figure in any of the algorithms

- what is a "safe strategy" regarding pruning PDBs?

- Figure 1: I think it's unfortunate to use S for size in contrast to before
where S denotes sets of variables/patterns

- "the formula used in our planner is revised": ? was it wrong before? is it
different than before? what is it?

- "if there is excess time and memory": this cannot happen since the
computation stops if either time or memory is exhausted

- the idea of the split domains in IPC'18 was that researchers can choose the
formulation that suits them better, but not to use both

- I don't understand the rationale for "GAMER-Style Comparison": if the
intention is to only compare that component of the overall planner (which is a
very good intention! this should be done for *all* components), then these
components should be evaluated in a *stand-alone* way, but not within the
planners that comprise so many other components which potentially impact and
falsify that comparison. Note that this is a general pattern for a good
experimental evaluation: isolate components and evaluate their impact
individually before combining them.

---

### Official Review · AnonReviewer1 · 2020-08-03
**Unclear contribution, careless experimental design**

**Rating:** 4
**Confidence:** 4

**Review:**

The paper presents a number of improvements over the Complementary pattern databases strategy presented in (Franco et al.2017)
and its IPC2018 instantiations CPC1 and CPC2. The implementation of the proposed improvements is dubbed CPC0 (which introduces
some confusion, I must say, for something which is meant to be _an evolution_ of CPC1).

I have the following concerns with the submission as it stands now, which I summarize here and elaborate below:

* First, the presentation of the contributions is not clear enough, and it is often hard to tell what is previous research and what is the proposed improvement.
The stated objective of the paper is "is to refine the PDB construction mechanism of CPC1 by optimizing the algorithms and the evaluator as well as better integrating
all components", but these optimizations and "integration" should be _concretly and precisely_ stated, described and motivated somewhere.
Take for instance the "GAMER-Style" subsection: it is not at all clear to me what parts of the algorithm come from (Kissmann and Edelkamp 2011), and what parts are novel.
Similarly happens for the bin-packing strategy described in the next subsection: what part of Algorithm 2 is from (Moraru et al. 2019) and what is novel?
Is the shuffle in line 11 the only novel contribution?


* The writing of the paper could be improved. In particular, the pseudocode used to describe the three presented algorithms is too informal,
with undefined variables.


* The proposed improvements are evaluated empirically over the IPC18 optimal track benchmarks, but although overall there
seems to be some improvement in terms of coverage, the design of the experiments (unnecessarily?) gives different time limits to the PDB construction phase,
which is the key thing being evaluated here (15 minutes to CPC1, 18 minutes to CPC0). I am a bit puzzled here as to why would it make sense to change the
experimental conditions here, but unless I'm missing something, to me this renders all subsequent comparison meaningless.
Also importantly, the experiments give a comparison between CPC1 and the novel CPC0 that incorporates all proposed modifications,
but there's no way of understanding which modifications account for which improvements - for that, some sort of ablation study would be required.


Overall, it is not clear to me what exactly is the conceptual contribution of the submission, and the empirical results
do not seem rigorous enough as to be considered a solid contribution in themselves.
I am therefore advocating for rejection.



Minor issues & typos:
* The definition of planning tasks in the Background section could and should be independent of the Fast Downward planner.
* "conjected" --> conjectured?
* "In CPC1, the perimeter PDB is constructed prior to the bin packing". What is the perimeter PDB?
* "In our implementation, the remaining candidate variables are shuffled before the partial run to avoid the effect of ordering (line 6)".
What is exactly meant by "the effect of ordering"?
* "Both bin-packing algorithms are implemented with more efficient data structures in CPC0." - What data structures are those?

---

### Comment · Program_Chairs · 2020-09-14
**Final Decision: Reject**

Dear authors,

We are sorry to inform you that we reject your submission to the HSDIP workshop. The reviewers agree that the contribution of the paper is not clear enough and the evaluation could be substantially improved. We hope that if you plan to submit your work to another venue the reviews allow you to substantially improve the paper.

Best,
The HSDIP'20 team

---

### Decision · Program_Chairs · 2020-09-30

Reject